# Divine Omnipotence, Divine Sovereignty and Moral Constraints on the Prevention of Evil: A Reply to Sterba

**Eric Reitan**

Department of Philosophy, Oklahoma State University, Stillwater, OK 74078, USA; eric.reitan@okstate.edu

**Abstract:** In *Is a Good God Logically Possible?*, James Sterba uses the analogy of a just political state to develop evil-prevention principles he thinks a good God would follow. With the assumption that God is omnipotent, these principles entail that God would never permit free agents to bring about horrendous evil. But free agents routinely succeed in doing so: entailing a logical incompatibility between the world's evils and the existence of a good, omnipotent God. I challenge this conclusion by sketching two ways divine omnipotence arguably entails that God would face moral constraints on the prevention of moral evil that human agents and political states do not. If my account is sound, God would be morally precluded from functioning as a sovereign governing authority in the manner of just political states. If this is correct, Sterba's arguments might be taken to show, not that there is a contradiction between the world's evil and the existence of a good, almighty God, but that there is a contradiction between the world's evil and the common theistic belief that such a God is the sovereign ruler of the world.

**Keywords:** problem of evil; divine omnipotence; divine goodness; divine sovereignty; theodicy; teleological and deontological ethics

## 1. Introduction

In *Is a Good God Logically Possible?*, James Sterba invokes the resources of moral and political philosophy to develop a new version of the logical argument from evil. He aims to show that a good God's existence is incompatible with "the degree and amount of evil that actually exists in our world" (Sterba 2019, p. 1) by formulating evil-prevention principles a good God would follow: principles drawn from the Pauline Principle (the principle that we should not do evil that good may come of it) and designed to capture our understanding of how morally good individuals and just political states would operate to prevent evil. His ultimate goal is to show that, given these principles, the degree and amount of evil in the actual world is greater than what a good God would allow to exist were that God all-powerful.

This work is important for how it centers moral philosophy in our reflection on the problem of evil. Sterba's work invites serious engagement with the question of what it means for God to be good. But while there is much of value in his attempt to show that a good God would not, if He could do otherwise, permit the degree and amount of evil in the world, I think his project falls short of demonstrating that there is a logical inconsistency between the evils that exist in this world and the existence of a good and omnipotent God.

In this essay, I will argue that there is a plausible account of God's goodness—an account reliant on the very Pauline Principle Sterba invokes—according to which God's power places moral constraints on God not imposed on those with less power. Unless Sterba can show this account to be untenable, cleaving to this account provides an escape from Sterba's conclusion.

But cleaving to this account has costs. Those who do so must renounce the common (not universal) theistic doctrine of divine sovereignty: the view that God is the sovereign ruler of the world. This doctrine lurks as an unstated assumption of Sterba's argument.

More precisely, in positing the just political state as a template for understanding how a good God would behave in relation to the world, Sterba is positing *both* (a) that a God who occupied such a role would be bound by principles of justice like those that define a just political state, and (b) that God, if existent, would occupy such a role. My account of divine goodness challenges (b) on the grounds that divine omnipotence imposes unique moral obstacles to taking up such a role. Theists who embrace my account thus preserve the compatibility of this world's evils with God's existence by holding that it is morally wrong for God to adopt such a role, and that this is why God allows evils that, were God occupying such a role, it would be impermissible for God to allow.

If this move offers the only effective response to Sterba's arguments, those arguments show something significant: they show that even if there is no contradiction between the amount and degree of evil in the world and the existence of a God who is perfectly good and all-powerful, a contradiction emerges when one adds the further doctrine (embraced by many theists) that God is the sovereign ruler of the world. Hence, unless Sterba's argument can be challenged in terms different from the ones proposed here, theists would be forced by the strength of Sterba's arguments to abandon the doctrine of divine sovereignty.

## 2. Sterba on the Free Will Defense

Sterba begins his version of the logical argument from evil by arguing that there is no successful Free Will Defense, because in the actual world there exist moral evils whose overall effect is to reduce what he calls "significant freedom", meaning the freedom "a just state would want to protect since that would fairly secure each person's fundamental interests" (Sterba 2019, p. 12). His claim is that one cannot invoke significant freedom's value as a reason for God permitting the world's moral evils, given that many of these evils reduce rather than increase the significant freedom in the world. His argument then considers other theodicies which appeal to other goods besides significant freedom, most notably the good of soul-making, and he argues that these other theodicies share a common structure: they justify God's permission of evil at least in part on the grounds that God can make up for it later.

It is here that Sterba invokes the Pauline Principle—that one ought not to do evil that good may come of it—which he takes to entail that one ought not to *permit* evil that good may come of it. But he notes that the Pauline Principle is not absolute, and that as such, there may be conditions under which one is justified in permitting evil that good may come of it. Most of the rest of the book aims to show that, with respect to the "significant and especially horrendous evil consequences of immoral actions" (Sterba 2019, p. 184)—what I will hereafter simply refer to as "horrors"—no such justification exists for God permitting rather than preventing them.

To begin my critique, I need to start with the kind of freedom whose optimization Sterba takes to be at the heart of the Free Will Defense: significant freedom. The first thing to note here is that Sterba deliberately attaches to this term a different meaning from the one Plantinga attaches to the term in the latter's development of the Free Will Defense. While Sterba, as noted above, defines "significant freedom" in relation to what a just political state would want to protect, Plantinga (1974, p. 30) uses "significant freedom" to name the freedom to pursue or refrain from "morally significant actions", which are actions that it would be morally right or wrong for the agent to perform. For Plantinga, the underlying presumption is that retaining the freedom to choose between moral good and moral evil—and hence retaining the capacity to choose moral evil—has a second-order positive value that God would want to protect even if agents who choose evil thereby bring about first-order negative values. Sterba's understanding of significant freedom does not in any obvious way preserve this presumption about the second-order values God would want to preserve, presumably because Sterba disagrees with the judgment that God would value this second-order value to the extent that Plantinga presumes. As such, given the different meanings attached to "significant freedom", it is not immediately apparent that the version of the Free Will Defense Sterba critiques is identical to Plantinga's.

A more significant concern for my purposes is this: Sterba's account of significant freedom conceives it in terms of *what a just political state would want to protect*, but my critique of Sterba involves arguing that God may be morally obligated to respect expressions of freedom that just political states would *not* be obliged to protect. As such, Sterba's formulation and critique of the Free Will Defense imbeds, within his understanding of the freedom God ought to optimize, the very assumption about how a good God would behave that I want to challenge. So long as the crucial assumption is thus buried within the terminology of "significant freedom", it is difficult to formulate and discuss the critique I want to develop here; one that questions whether the principles of intervention in people's affairs that govern political state are a good model for understanding what would govern the interventions of a good God.

Fortunately, there is an alternative construal of significant freedom that, when paired with some related concepts I am about to introduce, can be used to formulate the same basic critique of the Free Will Defense that Sterba wishes to push while disentangling the key moral premise from his definition of significant freedom. Specifically, let us construe "significant freedom" as the freedom to perform actions that have effects for good or ill.[1] The "effects for good or ill" might be intrinsic to the actions themselves—that is, the acts might bring good or bad into the world by virtue of being *intrinsically* good or bad—or the effects might be in the outcomes of the acts. If we construe significant freedom thusly, we can define *freedom-constraining actions* as active interventions in the exercise of free choices that either (a) prevent others from performing certain kinds of actions at all or (b) block those actions from having the kinds of effects they would have (in accord with natural causal laws) absent intervention. Further, we can define *freedom-policing actions* as freedom-constraining actions performed by a just state or its agents according to the principles governing a just state.

Using this terminology, we can reformulate Sterba's critique of the Free Will Defense as pushing the following point: a just political state does the most to protect the significant freedoms that a just state *should* protect when it engages in freedom-policing actions aimed at preventing horrors either by (a) preventing agents from being able to perform horror-producing actions or (b) mitigating the effects of those horror-producing actions such that they fall short of producing horrors. Sterba notes that the successful commission of horrors does more to reduce significant freedom (not only in his sense but also in mine) than would a carefully tailored policy of freedom-policing that targets horror-producing acts. Many of the objections theists pose to God engaging in freedom-policing acts fail to recognize the possibility of (b). So, for example, theists worry that if God polices horrors, God's omnipotence will entail that no one can ever successfully commit a horror. But if no one can successfully commit horrors, then no human agent would be motivated to try to stop those trying to commit horrors. In effect, people would see their choices in response to the evil plans of others as irrelevant, because those choices are rendered insignificant (they no longer effect the world for good or ill, since God will secure the good no matter what they do). And this, theists argue, is a serious cost. But Sterba rightly responds by noting that a carefully tailored freedom-policing policy that focuses on mitigating the effects of horror-producing acts would not have this cost. He asks us to envision God allowing the actualization of *some* of the negative costs of horror attempts if human agents were available to intervene but did not make the attempt, and God helping out to ensure none of the negative costs of horror attempts are realized when human agents do act to stop the horror.

Sterba's argument, formulated in my terms, is that such a carefully tailored freedom-policing policy, with the effect of eliminating horror from the world, would do more to secure significant freedom than would a hands-off policy. Thus, God's concern for significant freedom cannot explain the world as it is, given the amount of horror that exists. And so, he concludes, there is no successful Free Will Defense.

### 3. Teleological and Deontological Formulation of the Free Will Defense

My critique of Sterba's argument rests on what I take to be a failure to recognize an important application of the Pauline Principle; a failure that impacts his initial discussion of the Free Will Defense and undercuts the decisiveness of his critique of it. In overview, the problem is this: Sterba construes the Free Will Defense teleologically rather than deontologically, and this construal can be explained by the fact that he sees the Pauline Principle as posing a problem for the justification of *permitting the evil outcomes of misused freedom*. But the Pauline Principle might be invoked at what we could call an earlier place, to pose a problem for the justification of *freedom-constraining acts*.

Let me begin by sketching out in general terms the distinction I have in mind between teleological and deontological understandings of the Free Will Defense.[2] By the former, I mean a formulation of the Free Will Defense that takes significant freedom to be one important good (among others) that good moral agents ought to try to promote as much as possible in the world. I should note here that such a teleological approach need not suppose that only the *consequences* of an action are relevant to the determination of its moral status.[3] A teleological approach could hold that acts possess, at least sometimes, an *intrinsic value* that makes them good or bad in themselves apart from their consequences. What is crucial for a teleological approach is that it construes the intrinsic moral character of an act as one good or evil that the act brings about among others. Prescriptions are then arrived at through some kind of holistic assessment of all the good and bad that acts produce. Generally speaking, teleological approaches to morality take it that an act is morally right if it does the most good; that is, it does the best job, among the available alternatives, of promoting the good and limiting the bad (however these things are understood). It is also worth noting that while the most famous teleological theory in this sense, utilitarianism, equates good and bad with pleasure and pain, other teleological theories—such as G.E. Moore's—acknowledge a plurality of goods ([Moore 1922](), pp. 146, 224–25).

A teleological approach to the Free Will Defense would presumably see the possession of significant freedom as a good to be promoted, and might also view freedom-constraining acts as intrinsically bad. But if one thinks that a freedom-constraining act has an intrinsic badness apart from its consequences, that intrinsic badness would be treated as one mark against it. A freedom-policing act carried out in terms of well-designed policies might, despite being freedom-constraining, have the effect of reducing the total number of freedom-constraining acts in the world. If so, then this one freedom-constraining act eliminates more instances of the badness intrinsic to freedom-constraining acts than it brings about. On a teleological approach, if all else were equal, this would be sufficient to render the freedom-policing act morally right despite its intrinsic badness. If, furthermore, one took into account the positive value of increased significant freedom resulting from the reduction of freedom-constraining acts, the case for the justifiability of the freedom-policing act would be strong despite the act's intrinsic badness: so strong there would have to be extensive negative consequences in order to judge it wrong.

But in a deontological approach, the intrinsic moral character of an act is *directly prescriptive*. For the deontologist, the intrinsic moral evil of an act is not just one value to be taken into account alongside other values in order to arrive at a determination of the act's moral status. Instead, this intrinsic moral evil is better construed as an intrinsic *wrongness*: the act is of a kind that one *absolutely* or *prima facie* ought not to do, apart from any consideration of the total value produced by the act.

The "prima facie" qualifier is intended to indicate that, at least for many deontological theories, at least some intrinsically evil acts can be justified. While some acts are, perhaps, of a kind that is never permissible (they are *absolutely* wrong), others are of a kind that may be permissible to perform with the right sort of justification: they are *prima facie* wrong. But from a deontological perspective, the justification of prima facie wrong acts does not reduce to the kind of weighing of goods and evils characteristic of teleological approaches: to justify an intrinsically wrong or evil act, for deontologists, it is insufficient to show that the evil intrinsic to the act is outweighed by the goods it generates. This is what the Pauline

Principle, under one clear interpretation, is trying to say. "Do not do evil that good may come of it" entails that an intrinsically evil act does not become permissible just because it produces more good overall.

So, what can justify an act that is intrinsically evil in the deontological sense? Given the range and diversity of deontological theories, I cannot provide a brief answer that is fully satisfying. So, at the risk of oversimplification, I will content myself with a model of justification drawn from W.D. Ross (1930), who first invoked the "prima facie" epithet to qualify moral duties and prohibitions. For Ross, to say an act is prima facie impermissible is to say it is of a kind that would render it actually impermissible were there nothing else morally relevant to be said about it; more precisely, if it were not also, at the same time, of *another* morally relevant kind. On this view, actions acquire their initial moral standing—as prima facie duties (requirements and prohibitions)—by virtue of being of a particular kind. But specific actions can be of more than one kind at once, opening up the possibility that they can simultaneously be of an impermissible and obligatory kind. When that happens, the respective moral duties need to be weighed against one another (for Ross, by an appeal to moral intuitions). A prima facie impermissible act, then, would not be justified by the total value of what it produces but, instead, be the fact that (i) it is not only of the prima facie impermissible kind but also, at the same time, of a prima facie obligatory kind; and (ii) an assessment of the competing duties renders the judgment that the prima facie obligation is more pressing (See Ross 1930, pp. 19–20).

So, for example, I may have a prima facie duty to promote and protect the welfare of my child, and I may have a prima facie duty to respect Bob's freedom. But if Bob is harming my child and I restrain Bob to stop this harm, my action is simultaneously of a prima facie obligatory and prohibited kind. Based on moral intuitions, the duty to protect my child is weightier than my duty not to constrain Bob's freedom, thus justifying the latter despite its prima facie wrongness.

Given the distinction between teleological and deontological approaches to understanding moral prohibitions, permissions and obligations, we can see that a Free Will Defense could be of both teleological and deontological formulations. Under the former, freedom would be construed as inherently valuable, and freedom-constraining acts are problematic because they eliminate something of value. A teleologist might concede that in addition to eliminating something valuable (the freedom of those constrained), freedom-constraining acts are inherently bad. But on a teleological approach, a freedom-constraining act would still be justified when the good produced exceeded the act's total evil (counting both evil outcomes and intrinsic evil). While it may be difficult to compare different kinds of goods, there are cases where wicked agents act to constrain the freedom of others. In those cases, a teleological approach would justify constraining the wicked agents' freedom based on the value of freedom itself, since the wicked agents' lost freedom is offset by the overall increase in freedom that results, and the badness of constraining the wicked agents' freedom is offset by preventing the badness of the wicked agents' freedom-constraining acts.

It is this way of thinking about the Free Will Defense that leads Sterba to conclude that it fails as a response to the problem of evil. As he sees it, the world is full of cases in which wicked agents commit horrors that truncate freedom far more than would an adequately constrained divine intervention. As a case study, he examines the brutal murder of Matthew Shepard. He argues that "there was no way that failing to prevent Matthew Shepard's murder could have been justified in terms of a gain in significant freedom when compared to the loss of significant freedom that resulted from the murder" (Sterba 2019, p. 23). Sterba concludes that "if God is justified in permitting such moral evils, it has to be on grounds other than freedom because an assessment of the freedoms that are at stake would require God to act preventively to secure a morally defensible distribution of freedom . . . " (Sterba 2019, pp. 23–24).

Similarly, Sterba argues that a just political state would be "committed to restricting the far less significant freedoms of would-be aggressors in order to secure the far more significant freedoms of their would-be victims" (Sterba 2019, p. 29). Here, as in the

discussion of Matthew Shepard, Sterba challenges the idea that God's respect for freedom would preclude divine freedom-constraining acts. Instead, God's respect for freedom would demand such acts because they would produce more significant freedom overall.

This is clearly a teleological approach to thinking about the Free Will Defense: Sterba argues that a just God who cared about significant freedom could do more to promote it by engaging in freedom-policing than by refraining from freedom-constraining acts. But what about a deontological version of the Free Will Defense? On a deontological approach, acts of respecting and constraining freedom have an intrinsic moral character, and this intrinsic moral character is not just one value to weigh up among others. Instead, it grounds a moral status independent of this total assessment of the resultant value.

The simplest version of such a deontological Free Will Defense would regard the duty to respect significant freedom as absolute. So construed, God would be morally precluded from engaging in freedom-constraining acts no matter how much freedom is thereby lost when immoral agents ignore this absolute principle. God's moral perfection would thus be construed in terms of obedience to an unconditional rule prohibiting even a nuanced policy of freedom-policing that increases significant freedom by constraining those who would stifle it.

While this strong deontological formulation of the Free Will Defense would reconcile the world's moral evil with the existence of a God who is omnipotent and perfectly good *in the stipulated sense*, the stipulated sense is implausible. Most have the strong intuition that a duty to respect significant freedom is at best a prima facie one, not absolute, and that we are justified in violating the significant freedom of agents engaged in or about to engage in horrors. Perhaps this is because we see ourselves as having a prima facie duty to care for others' welfare, which in turn implies a prima facie duty to prevent horrors when we can do so easily and without significant personal cost. Following Ross's model for justifying prima facie impermissible acts, we would say the act of constraining the significant freedom of wicked agents to prevent them from committing horrors is simultaneously of a prima facie impermissible kind (a freedom-constraining act) and obligatory kind (a horror-preventing act). Given the intuition that the latter duty is weightier, constraining the freedom of horror perpetrators is justified.

If we apply these moral intuitions to God, God would have a duty to refrain from freedom-constraining acts except to prevent the commission of horrors. But Sterba rightly notes that in the actual world, agents routinely succeed in carrying out horrors.

As such, shifting from a teleological to a plausible deontological construal of the Free Will Defense does not by itself save the Free Will Defense from the kinds of concerns Sterba raises. But recognizing the possibility of a deontological construal nevertheless opens a door that Sterba believes he has closed. The reason is this: even if we grant that, in relation to human agents, a prima facie prohibition against freedom-constraining acts is routinely overridden, one might suppose that the justifying conditions for such freedom violations have something to do with unique features of the human condition that do not apply to God. If that is so, then a weak deontological formulation of the Free Will Defense might yet succeed.

## 4. Applying the Pauline Principle at an Earlier Place

Given the importance Sterba places on the Pauline Principle—a principle that firmly endorses a deontological approach to thinking about the relationship between the morality of actions and the good (or evil-prevention) that they produce—it may be surprising that when he tackles the Free Will Defense, his interpretation is so thoroughly teleological rather than deontological. But this is less surprising when we look more closely at how he applies the Pauline Principle. What I will argue here is that even though Sterba's moral approach encompasses deontological concerns, as evidenced by his invocation of the Pauline Principle, the *place* in his moral thinking at which he invokes that principle leads him to an essentially teleological construal of the Free Will Defense. And as such, he misses a more explicitly deontological construal that *invokes the Pauline Principle at an earlier place*.

To see why, let us turn to Sterba's use of the Pauline Principle. The Pauline Principle, as Sterba formulates it, states that "we should never do evil that good may come of it" (Sterba 2019, p. 49). When Sterba introduces this principle, he conceives it not as an absolute principle but, rather, as one that admits of exceptions: specifically when the evil at issue is (1) trivial or (2) easily reparable or (3) "the only way to prevent a far greater harm to innocent people" (Sterba 2019, pp. 49–50). In my terms from the discussion above, Sterba holds that *permitting evil* is an intrinsically evil kind of act; hence, it is prima facie wrong and in need of justification. While the fact that good comes of it is never by itself a sufficient justification, (1)–(3) may provide the needed justification.

In applying this principle to his argument, Sterba's main focus is on *permitting significant and especially horrendous moral evil*: for ease of reference, what I will refer to as "permitting horror". Sterba's argument is that to permit horror is to do evil of a certain kind, and as such falls under the Pauline Principle that one may not do evil that good may come of it. But, if God exists and is all-powerful, God clearly permits horror. As such, the theist must account for God's permission of horror by showing that it falls under one or more of the exceptions to the Pauline Principle; that, despite being prima facie wrong, God's permission of horror is nevertheless justified. The mere fact that good may come of permitting horror is not enough to justify it. More is needed. And classic attempts to provide that "something more" are failures.

Given where it starts, this overall line of argument is compelling. The problem lies with where this line of reasoning starts. Specifically, the kind of divine act that Sterba identifies as a case of doing evil, and hence as in need of justification, is the divine act of *permitting horror*. And the divine act of permitting horror is best described as an act of omission.

I do not want to argue that omissions cannot be intrinsically evil and hence prima facie impermissible. I think they can. But I also think that when the act in question is an omission—when it is a case of *refraining* from what we might call "positive" action—it must meet a distinctive condition before we can say it is intrinsically evil and so a case of "doing evil". The condition is that the positive action one is refraining *from* is not *itself* intrinsically evil. If one omits a course of positive action, P, because P is intrinsically morally wrong, then the omission cannot be intrinsically morally wrong unless (a) P is only prima facie intrinsically wrong, (b) the prima facie case against it has been overridden by the circumstances such that P is justified and (c) one persists in refraining from P. Put simply, an omission cannot be intrinsically wrong and so a case of "doing evil" if the alternative to omission is the commission of an act P, where P is an intrinsic evil whose commission *is not justified*. In such cases, the Pauline Principle applies to the commission of P and so cannot apply to its omission.

When one applies the Pauline Principle to an omission, but one has failed to sufficiently consider that the alternative to omission may be the commission of an act ruled out by the Pauline Principle, I will refer this as starting in the wrong place with respect to the application of the Pauline Principle. My suggestion here is that Sterba starts in the wrong place when it comes to applying the Pauline Principle to divine freedom-constraining acts aimed at preventing horror.

More precisely, the proposal I want to consider is this: freedom-constraining acts should be construed as intrinsic evils and hence as prima facie impermissible acts in need of justification. Absent justification, they are instances of doing evil, and so prohibited by the Pauline Principle.

Construed in this way, freedom-constraining acts aimed at preventing horror are intrinsically evil insofar as they are freedom-constraining, but potentially justified insofar as they prevent horrors. The ill effects of omitting the freedom-constraining act (the resultant horrors) function as a potential justification for the otherwise prohibited evil of constraining freedom. Until that potential justification is evaluated and found satisfactory, one cannot say that the omission is itself intrinsically evil. First, we must evaluate the justificatory power of the fact that omission permits horror. Only if it succeeds as a justification can we

then label the omission that permits horror as *the evil act of permitting horror*: an evil in need of justification.

What is crucial here is the sharp difference between the kind of moral reasoning required to determine whether "failing to constrain freedom would permit horror" justifies constraining freedom, when such constraint is prima facie wrong, and the kind of moral reasoning required to determine whether permitting horror, *posited* to be prima facie wrong, can be justified by the good that may come of it. The success of a project pursuing the latter, insofar as it *assumes* the success of a project pursuing the former, cannot serve as a case for the former. Thus, if one pursues the latter project without first pursuing the former, one has applied the Pauline Principle in the wrong place. This, I argue, is the mistake Sterba makes.

## 5. Unlimited Redemptive Power

This error opens Sterba up to two objections, both of which feature God's power in different ways. The first objection I will consider appeals to God's redemptive power. By redemptive power, I mean the power to mitigate or eliminate the negative impact of involvement with evil. We might say that evils have the power to diminish or even destroy the positive value and meaning of a person's life. Following Marilyn McCord Adams, we can attach the label "horrendous evil" or "horror" to evil that prima face strips life of positive worth or, put otherwise, "gives one reason prima facie to doubt whether one's life could . . . be a great good to one on the whole" (Adams 1990, p. 211). By "participation" in such evil, Adams means both the doing and the suffering of it, since horrors have the power to strip worth in both cases, although in different ways. To say horror strips life of worth *prima facie* leaves open the possibility that some action or occurrence could restore worth to horror participants.

Such restoration is what I mean by "redemption". On this understanding, the redemption of evil is always with respect to some person caught up in evil. When the redemption is achieved through some third party's actions, we might call such action *a redemptive intervention*. A redemptive intervention is *partial* if, when deployed in response to evil, a person's life acquires more value than would have been the case if the person had been caught up in the evil but not been the object of the redemptive intervention. A redemptive intervention is *complete* or total if, when deployed in response to evil, the person's life retains all the value it would have had absent being caught up in the evil in the first place.

Adams, in her work, distinguishes between two distinct ways that God might act to restore meaning and value to the lives of those caught up in horror (Adams 1990, pp. 218–20). Both would qualify as redemptive interventions in the sense defined here. The first way God might act to overcome horror is by engulfing it through the bestowal of the beatific vision; that is, the direct experience of God's presence and love. As Adams notes, in traditional theology the good of such direct experiential connection with God is of such extraordinary worth that it swamps all finite evils, so engulfing them that even what would seem an evil of insurmountable magnitude absent the beatific vision is, within the context of such an infinite good, rendered trivial by comparison. In addition to engulfing horror, God could also act to defeat it. By this, Adams means the act of building up around the horror something of positive value such that the evil becomes an integral part of a greater good. By making it such that the horror becomes a component of a greater good, God thereby deprives the horror of its power to diminish the meaning and value of a life: because the horror is now an integral part of a greater good so valuable that the horror victim would not want to do without that good, even though its existence depends upon the horror. Adams argues that horror calls for being not merely engulfed but defeated. In her book *Christ and Horrors* (Adams 2006, pp. 53–79), she posits the Incarnation and Crucifixion of Christ as horror-defeating divine interventions insofar as these divine acts turn the emotional place of horror into the singular place in a human life where one can exist in solidarity with God at God's most accessibly human. This power to creatively intervene to defeat horror, combined with the infinite value of the beatific vision, entails for Adams that God's power to redeem horror is essentially *unlimited*.



By contrast, horrors are precisely those evils that human agents are generally powerless to redeem. In some cases, horror victims may reach a place in this life where, on the whole, their lives have positive worth. But there is little human agents can do to guarantee this outcome, and many horror victims die with horror as the defining fact of their terrestrial lives. This means that if I am in a position to easily prevent someone, V, from enduring horror but fail to do so, I have chosen a path that precludes me from acting on the whole in a way that could be rightly described as "being good to V", even in a minimal sense.

What follows is that if I can easily prevent V from enduring horror by constraining the free choices of a wicked agent, my prima facie obligation not to constrain freedom clashes with my prima facie duty to be minimally good to V. And that would justify me in constraining freedom. Given the magnitude of what is being prevented, we tend to think that even a serious constraint on freedom would be justified.

Arguably, however, the strength of this justification for constraining freedom is a function of the extent of one's capacity to redeem the evils that result from misused freedom, precisely because, with significant capacity to redeem those evils, our capacity to be minimally good towards the victims of horror in ways that *do not* involve constraining freedom is retained. Put simply, the more power I have to redeem the evils resulting from wicked acts, the weaker my justification for constraining the agents of those acts.

Suppose I am a grade school teacher supervising children at recess, and I see an altercation among the children in which a bully starts to act aggressively towards another child. At what moment do I intervene? Immediately? Or do I give the kids the space to exercise their freedom until the bully's actions threaten harms beyond what it is in my power to repair? It may be hard to get a clear intuition here, since the human capacity to repair harms to others is so limited (emotional harms might be particularly hard to address). Given how limited my power to repair harms is and my lack of foreknowledge of the altercation's trajectory, I may jump in quickly rather than risk harms beyond what I can repair.

But if we suppose that God's capacity to repair harms is unlimited, the considerations that would justify freedom-constraining interventions might not merely be *less* compelling. They might vanish altogether, such that God is morally *precluded* from intervening precisely because God has the power to meet the obligation to be good to the victims of moral evil in a different way: by engulfing and defeating the evils they endure.

Sterba (2019, pp. 141–51) does consider whether divine redemptive activity might justify God's permitting horrors. When he speaks explicitly about redemption, he uses the term in a much narrower sense than I have defined it here: as activity aimed at assuaging the guilt and achieving the moral reform of the agents of evil. But elsewhere (Sterba 2019, pp. 36–44), he considers how friendship with God might repair the damage done to victims of horror, and how Jesus suffering along with us might be a balm in the midst of that suffering. As such, he considers the other dimensions of what I mean here by the redemption of evil: engulfing and defeating evil. And he argues, plausibly, in each of these cases, that it would be better had the evil never been done in the first place than that it be done and then redeemed. So, if redemption is invoked to justify God's permitting horrors, it fails.

But what I am arguing here is not that God's capacity to redeem horrors justifies God in permitting them. What I am arguing, instead, is that God's justification for constraining freedom—namely that it is necessary to prevent horrors—is undercut by the fact that God, by virtue of an unlimited divine power to redeem evils, has an alternative means of guaranteeing that horror victims have lives whose value is undiminished by horror. God's unlimited capacity to redeem horrors strips God of the basis for justifying the freedom-constraining acts required to prevent horror in the first place. Thus, the question of whether God is justified in preventing horror never arises, because the horror-preventing act is precluded by its intrinsic wrongness, a wrongness not overcome by a sufficiently powerful justification. Since that wrongness is not overcome, and since it is wrong for God to do

evil that good may come of it, it is wrong for God to prevent horrors; even though it is not wrong for us.

In other words, Sterba misses a way in which the theist might go about accounting for God's apparent permission of horror; a way that he misses because he invokes the Pauline Principle in the wrong place. If one invokes the Pauline Principle where Sterba does, to label God's permitting horrors as prima facie intrinsically wrong and in need of justification, then God's redemption of horror will be treated as a potential justification for allowing it, and a rather poor justification, as Sterba argues. But before one can legitimately address God's redemptive power on these terms, one must first invoke the Pauline Principle where I propose: to label God's freedom-constraining acts as intrinsic evils in need of justification. Here, it is the fact that these acts prevent horrors that function as the purported justification, and God's redemptive power is introduced to account for why, in God's case, this justification is inadequate.

Sterba himself notes that greater power can deprive one of justifications for action that would be available to those with less power ([Sterba 2019](#), pp. 78–80). For Sterba, this comes up when he considers typical reasons why finite creatures such as ourselves might be justified in permitting evils to occur: because (rather routinely) we lack the causal power to prevent the evil without thereby also permitting a greater evil or the loss of a greater good. But God's unlimited causal power, Sterba argues, deprives God of this kind of justification.

The argument I propose here follows a similar principle: our performing the intrinsically evil act of constraining freedom is justified (rather routinely) by the fact that such acts are the only way available to us of being good to the victims of horror and preventing the long-term evil effects of horrors. But that sort of justification for doing something intrinsically evil is unavailable to God if God can fully meet His obligations to the victims of horror through the exercise of his limitless power to redeem horror.

Put simply, it is not in my power to fully respect the freedom of every moral agent while also fully expressing care for the welfare of every person whose welfare I have the power to impact. This is so because it is not in my power to redeem the damage to human lives that results from some misuses of freedom. So, if I respect freedom to the point of not intervening in those misuses, I fall short of respecting human welfare. But it is in God's power *both* to respect fully the freedom of every moral agent *and* to respect fully the welfare of every person (ensuring that every person has a life with as much value and meaning as it is possible for a human life to have). This is possible because of *God's infinite capacity to redeem* the lives of those caught up in even the most horrific moral evils. That God can effectively erase the evil from the world after it has occurred by fully redeeming it (something none of us can do) could arguably entail that preventing the evil from happening in the first place no longer functions as a sufficient justification for violating the prima facie prohibition against freedom-constraining acts.

In short, Sterba explores whether God's capacity to and intention to redeem all evil can justify His doing the evil act (of omission) that permits the evil to occur. My question is whether God's capacity to and intention to redeem all evil can undercut a justification for His doing the evil act (of commission) of constraining freedom. Even if Sterba has a sound argument against the view that God's redemptive power justifies God in permitting evil (I think he does), it does not follow that God's redemptive power cannot play an important role in establishing an effective theodicy. This is because it may be the case that what would justify us (who lack God's redemptive power) in constraining the freedom of other agents cannot justify God (who has that redemptive power) in doing likewise.

Applying the Pauline Principle at an earlier place, then, offers the basis for a deontological Free Will Theodicy, one that sees acts of constraining freedom (once there exist beings who possess it and whose nature inclines them to use and value it) as intrinsically evil and so in need of justification. While humans are routinely justified in constraining the freedom of others—at least when their actions rise to a sufficiently serious level such that they are using their freedom to commit horrors that harm both the welfare and freedom of others—this is because they lack an alternative means of showing the concern for human

welfare that morality demands. But God, by virtue of possessing unlimited redemptive power, has such an alternative means and so lacks the justification we possess to set aside the prima facie duty not to constrain freedom. Lacking such a justification, God's hands are tied: *morally* tied by a deontological constraint that, by virtue of our limited power, we do not possess.

The moral perspective I propose here can be summed up in terms of the following four moral claims:

1. It is prima face impermissible to constrain a person's significant freedom, implying that the act of constraining the significant freedom of horror perpetrators is prima facie impermissible.

2. If the prima facie impermissibility of constraining the significant freedom of horror perpetrators is not overcome by a sufficiently compelling justification, then constraining the significant freedom of horror perpetrators in order to prevent the evil they would otherwise do amounts to a violation of the Pauline Principle. It would be an impermissible instance of doing evil that good might come of it.

3. The prima facie duty to show minimal concern for the good of horror victims would be a sufficient justification for performing the prima facie impermissible act of constraining the significant freedom of horror perpetrators *unless* the agent had available to them an alternative way to be just as good to the victims of horror; a way that did not come at the cost of violating significant freedom.

4. God's unlimited capacity to redeem evil entails that God always has available a way to be just as good to the victims of horror as God would be were God to prevent the horror; a way that does not come at the cost of violating significant freedom.

If 1–4 are true, then God's permitting horrors is not an instance of God violating the Pauline Principle but a consequence of God being morally constrained by the Pauline Principle: God is prohibited from doing the evil of constraining freedom even that the good of horror prevention may come of it. What justifies *us* in constraining freedom—the duty to be good to a horror's victims—cannot justify God in constraining freedom, because God's unlimited capacity to redeem horrors means God has another way to be just as good to a horror's victims.

Note here that my aim is not to argue that 1–4 are true. Rather, my aim is to point out that if 1–4 are true, Sterba's argument fails. And insofar as Sterba has failed to show 1–4 to be untenable, he has failed to demonstrate that the degree and amount of evil in the world is incompatible with the existence of a good and all-powerful God.

## 6. Unlimited Policing Power

In the previous section, I argued that before we can ask whether God permitting agents to perpetrate horror can be justified, we must ask whether God constraining the freedom of those agents can be justified by the horror thereby prevented. If we assume that constraining freedom is an intrinsic evil, the Pauline Principle entails that the good outcomes of constraining freedom are not by themselves sufficient to justify it. Nevertheless, constraining freedom might be justified as the only way to carry out the moral obligation to show proper concern for the good of a horror's prospective victims. And while that justification would be a powerful one for agents with limited power to redeem horror—and so would generally justify humans in constraining freedom as a means to prevent horror—it fails to provide God with a justification for constraining freedom if we assume that God has limitless power to redeem horror and is thus able to fully carry out the duty to be good to a horror's victims without constraining freedom.

In this section, I want to suggest a second way in which God's omnipotence could limit the justifications for constraining freedom available to God. In this case, however, I want to consider God's freedom-constraining activity explicitly in terms of Sterba's analogy to an ideally just political state. Sterba's assumption is that God would relate to the world in a manner analogous to such a state. And one of the key features of such a state is how it regulates the freedom of those who fall within its jurisdiction. My argument here is this:

the principles that would guide an ideally just political state are reasonable standards for assessing God, only if we assume that God legitimately (morally) may occupy the role of sovereign governing authority over the world.

A sovereign governing authority, unlike a private individual, exercises extensive power over the free choices of those who fall within the scope of the authority's rule. Such an authority establishes the limits on how freedom may be used and polices misuses of freedom. For this very reason, we tend to think that there are moral conditions that must be met before someone can assume this role. Put another way, the assumption of such a role, given that it involves the use of extensive power to significantly constrain the freedom of others, requires justification. To operate *as* a sovereign governing authority absent such a justification is to act wrongly.

An ideally just political state thus needs to be understood not merely substantively—in terms of the principles according to which it governs, including those it uses to make decisions about when and how police the exercise of freedom among its subjects and citizens—but also formally, in terms of the basis on which it assumes the role of sovereign governing authority in the first place. And arguably, this is so because exercising governing authority amounts to extensively and systematically regulating human choices within a community by imposing rules constraining freedom and policing obedience to those rules, and such systematic control over the lives of others is prima facie problematic even if the principles used to govern are themselves good ones. Put in terms of an analogy, if immensely powerful aliens (unburdened by anything like the Prime Directive of the *Star Trek* universe) were to come to Earth and assume control, erasing all elected governments and human laws and replacing them with their own, there is arguably a significant basis for moral complaint against these aliens *even if the principles by which they governed are sensible and just*.

But suppose, instead of a direct alien usurpation of governing authority, a single alien with extraordinary power, raised among us and bedecked in blue tights and a red cape, makes a commitment to intervene to stop every terrestrial villain who misuses their freedom to exploit or abuse others. Sterba invites us to consider such a superhero, and thinks we will agree that there is nothing objectionable about such a hero using their power to prevent such villainy. "In fact", he says, "inaction by the superheroes in such contexts is broadly condemned by virtually everyone ... ". (Sterba 2019, p. 19) He goes on to imagine that such uses of superpowers are not limited to "protecting people from serious assault" but extend to "protecting people from the significant evils of an unjust economic system, thereby securing people's freedom in that area of their lives", envisioning Robin Hood-like uses of superpowers to ensure equity in defiance of systemic forces at odds with equity (Sterba 2019, p. 20).

Arguably, if the power of the superhero is sufficiently limited, such interventions might still be welcomed without complaint. Even Superman is just one man, and his super-hearing has limits. Keeping the criminals in check is a full-time job even absent super-villains, and so we would not imagine that his efforts would result in one man systematically usurping the government and replacing the existing laws and policies and enforcement system with his own. But if we imagine Superman to be sufficiently powerful, then Superman doing everything within his power to prevent or correct individual and systemic evils would amount to Superman *becoming the de facto sovereign authority of the world*.

This is because, with sufficient power, Superman's interventions would amount to the creation of de facto public policies. If, according to Superman's astute sense of justice, actions of type X are wrong, then anyone who tries to perform actions of type X will be stopped regardless of whether actions of that type are against the laws laid down by the elected government. If, having read the best ethical reflections on economic policy, Superman uses his powers to police the decisions of corporate executives, stymying their efforts to maximize shareholder profits by exploiting workers and ignoring environmental health, it would mean the implementation of a new and different economic system than the one we currently have in place.

And this is something I could imagine many would object to, even if Superman is guided by sound principles. With sufficient power—the kind of power that generally in human affairs requires the coordinated activity and consent of many individuals—Superman could, hypothetically, assume the de facto role of sovereign governing authority of the world without the coordinated activity and consent of others. All it would take is a consistent commitment to constraining human freedom in accordance with a set of principles and the power to carry out that commitment with enough regularity that defiance will generally fail. In short, if we add to Superman's already extensive powers additional abilities that enable him to implement his vision of justice on the world, it would follow that Superman's freedom-constraining acts would become freedom-policing ones: they would be the principle-governed actions of a person who has assumed the role of supreme governing authority over the world.

Even absent the relevant level of power, there is some moral difficulty surrounding the activity of an individual intervening in free choices in the manner of a law-enforcement officer but without being officially appointed to that role. We call such individuals vigilantes, and the fact that their activities have a controversial or contested moral status even if the substance of what they do accords with our sense of justice highlights the moral significance we attach to the more formal dimension of the legitimation of freedom-policing actions. But the problem is clearly magnified if the individual has so much power that there is little difference between that individual protecting people as far as they can from the unjust effects of misused freedom, and a fleet of Kryponians arriving on Earth and announcing that they have assumed control.

Clearly, based on traditional theistic assumptions, God has enormously more power than that Kryptonian fleet. As such, if God were to do everything within the divine power to police misuses of freedom, God would—by virtue of divine unlimited power—thereby assume the role of sovereign governing authority over the world. In fact, even if God were to do a fraction of what God could do to police misuses of freedom, God would still assume that role. If God were to assume such a role, then Sterba offers some quite sensible principles for how God should govern. As Sterba notes, it might make sense for God to leave room for humans to have an impact on the world by ensuring that, if they are in a position to prevent evil but do not do so, things turn out less well than would have been the case had the human agents intervened, but not so badly as to result in horror.

But the substantive question of how God should govern assumes an affirmative answer to the question of whether it is morally acceptable, in the world as it is, for God to assume sovereign governing authority. But it seems a mistake, given Sterba's aim of establishing a logical contradiction between the existence of God and the degree and amount of moral evil that exists in the world, for Sterba simply to assume that there are no moral constraints against God adopting the role of sovereign governing authority over the world. And given divine omnipotence, the question of whether God should prevent all the serious moral evils that God can eliminate amounts to the question of whether God should become the sovereign governing authority over the world. If there are moral impediments to God assuming such a role, then the fact that the world does not look the way it would were God occupying and acting in accord with such a role falls short of a decisive logical case against God's existence.

To decide whether there are moral impediments to God assuming such a role, we must have two things: a clear account of what counts as the basis for legitimately holding and exercising such authority, and an account of the conditions in this world sufficient to determine whether this basis is in place. With respect to the former, a key question is to what extent *consent of the governed* is required, and in what form, for the adoption of the role of sovereign governing authority to be morally legitimate. Our political traditions certainly affirm the idea that some form of consent on the part of the governed is required before someone can assume a role which entails such far-reaching interventions in the exercise of significant freedom.

With respect to the latter, we must consider what it would look like for humanity to give God consent to rule, and whether humanity has to any significant extent in its history done so.

Of course, there are theists who would confidently claim that God is unique in not needing the consent of the governed in order to have a right to rule, but that confidence needs to be weighed against other considerations. Arguably, the kind of autonomous agents God created in fashioning humanity possess, by virtue of their nature, a right to play a role in who adopts the role of sovereign governing authority in their communal lives. Furthermore, since God created them as the kinds of beings who not only possess but value their freedom and autonomy, God thereby brought it about that there exist creatures who have a right to not be ruled by someone without some kind of collective consent.

With respect to the latter, even in societies that profess to desire to be ruled by God there are sufficient displays of human ego and pride and posturing to allow for the interpretation that these professions are insincere, at least on a scale large enough to warrant the judgment that humanity has given consent to God taking charge in the manner of a sovereign authority.

In any event, these are issues Sterba has not taken up, and unless and until Sterba does so, his case for a contradiction between the existence of a good, omnipotent God and the degree and amount of moral evil in the world remains inconclusive. Put simply, if God doing even a fraction as much as it is in God's power to do to prevent horror amounts to God de facto assuming the role of sovereign governing authority over the world, and if it would be morally impermissible for God to assume such a role absent the right kind of consent of the governed, then it would be wrong for God to do even a fraction as much as it is in God's power to do to prevent horrors. Furthermore, it would, arguably, be wrong even if doing so would result in a better overall balance of good and evil in the world, including the good of significant freedom and the evil of freedom-constraining acts. Thus, again, divine omnipotence may lead to God running afoul of deontological moral constraints that would not constrain the less powerful.

The argument here is not that this is the correct moral picture to adopt, only that it has some plausibility given our larger moral views on the conditions of legitimate authority to engage in freedom-policing. Hence, Sterba must tackle this moral picture and demonstrate why it fails before he can claim to have established a logical incompatibility between the degree and amount of moral evil in the world and the existence of a good and all-powerful God.

## 7. Summary in Terms of Sterba's Moral Evil Prevention Requirements

One useful way to summarize these objections to Sterba's argument is to reframe them in relation to the three "Moral Evil Prevention Requirements" (MEPRs) that Sterba thinks have not been met by God in the world. Sterba spells these requirements out as follows:

I.　　Prevent, rather than permit, significant and especially horrendous evil consequences of immoral actions without violating anyone's rights (a good to which we have a right), as needed, when that can be easily done.

II.　　Do not permit significant and especially horrendous evil consequences of immoral actions simply to provide other rational beings with goods they would morally prefer not to have.

III.　　Do not permit, rather than prevent, significant and especially horrendous evil consequences of immoral action on would-be victims (which would violate their rights) in order to provide them with goods to which they do not have a right, when there are countless morally unobjectionable ways of providing these goods. (Sterba 2019, p. 184)

Essentially, Sterba argues that if God exists and is good, God will follow MEPR I-III. But if God followed MEPR I-III and was all-powerful, there would be no horrendous evil. Since there is horrendous evil, there does not exist a God who is good and all-powerful.

With respect to MEPRs II-III, it should be clear that my proposed rationale for why God permits, rather than prevents, the horrendous evils we see in the world is neither to provide other rational beings with goods they would rather not have nor to provide goods to which they do not have a right. Instead, the proposed rationale is that God is morally constrained by deontological prohibitions against violating significant freedom.

According to the first argument, the constraint comes from a prima facie prohibition against violating freedom: one which is routinely overridden in the case of finite persons by the more pressing weight of competing duties, but which is not so overridden in God's case because God's unlimited capacity to redeem evil entails God can meet these other duties without violation of significant freedom. According to my second argument, the constraint comes from a prohibition against becoming the de facto governing authority of the world without the consent of the governed: a prohibition that given God's power, God would violate if God engaged in even a fraction of the horror prevention of which God is capable.

If God permits horrors because of such deontological constraints, God is not violating MEPR II or III. Hence, there is a potential account of God's permission of the horrors of the world that is not ruled out by these requirements.

With respect to MEPR I, the deontological perspective proposed here can be seen as either calling for a revision to MEPR I or a distinct interpretation of it. On the former approach, the deontological critic of Sterba would propose the following alternative:

> MEPR I*: Prevent, rather than permit, significant and especially horrendous evil consequences of immoral actions when that can be easily done without violating any active deontological moral duty (such as those imposed by persons' rights).

The argument of this paper is that there is a plausible moral perspective Sterba has not tackled: one which holds MEPR I*, and according to which there do exist active deontological moral duties that would preclude God from preventing horror, even in cases where no such duties obtain for persons with limited power. Where MEPR I* would require that finite persons prevent horror, the same is not the case for God.

Alternatively, with a sufficiently robust notion of the *correlativity* of rights and duties, one might interpret Sterba's formulation, MEPR I, as equivalent to MEPR I*: if God has a duty to respect the significant freedom of finite persons, one might say those persons have a correlative right—at least against God—for that freedom to be respected. In that case, the argument here would be that given God's unlimited capacity to redeem horrors and the way in which that strips God of the justification (generally available to humans) for violating the prima facie duty to respect significant freedom, the prima facie human right to exercise significant freedom is rendered absolute in relation to God and so entails rights against God that do not apply against other finite persons.

Whether one formulates the response as a revision of MEPR I or an interpretation of it, the conclusion is the same: a plausible moral perspective that Sterba has failed to consider entails that if God existed and were almighty and good, the horrors we see in the world might still obtain.

## 8. The Case against Divine Sovereignty

In the preceding, I have argued two things. First, I have defended the plausibility of the idea that the more power one has to *redeem* the evil consequences of misused freedom, the less those consequences can justify violating a prima facie prohibition against freedom-constraining acts. If God is all-powerful, then God arguably possesses an unlimited capacity to redeem evil and so is barred from all freedom-constraining actions, and so must permit horrors. That is, God is morally prohibited from policing human freedom in the ways that human political states—with their limited capacities to redeem horror—are not only permitted but morally required to do.

Second, the more power one has to prevent misuses of freedom (and the outcomes of such misuse), the more likely it is that doing everything in one's power to prevent "significant and especially horrendous consequences" of misused freedom amounts to

adopting the role of sovereign governing authority over humanity, and thus potentially running afoul of moral principles dictating the conditions under which one can rightly assume such a role. Given divine omnipotence, God *will* become the de facto governing authority of the world unless God does far, far less in terms of freedom-policing than God is capable of doing. In fact, even a tiny fraction of the power at God's disposal would, if implemented in the project of policing misuses of freedom, reflect a level of sovereign authority over the world that swamps what any elected human authorities could achieve. Hence, if there are moral principles that require the consent of the governed before someone may adopt the role of sovereign governing authority over the world, God may be morally precluded from exercising even a fraction of the policing power at God's disposal absent such consent. And it is at least arguable that human societies have only paid lip service to the idea of giving the rule of the world over to God, and there has never been anything like the consent of the governed being morally required for God to assume that role.

Both of these arguments converge on the conclusion that it is wrong for God to assume the role of sovereign governing authority over humanity. The second argument does so directly, but the first argument does so indirectly: if God is barred from constraining the freedom of others by virtue of God's unlimited power to "make right" the consequences of misused freedom, then God is morally precluded from "policing" misuses of freedom. Insofar as the role of sovereign governing authority presupposes a right and duty to police misuses of freedom, it follows that God is morally precluded from assuming that role. In fact, the first argument is more powerful than the second: if it succeeds, it may be morally wrong for God to adopt the kind of role in human affairs that a just political state adopts, even if the concerns raised in the second argument are adequately addressed. Even if some properly conceived mechanism for securing the consent of the governed in relation to God is implemented, it may be wrong for God to assume control.

This point is significant because of the extent to which theistic traditions have held that God *is* the governing authority over the world: a point especially prominent in Calvin's theology and that of those who follow him. The following is characteristic of Calvin's view:

> For [God] is accounted omnipotent, not because he is able to act, yet sits down in idleness, or continues by a general instinct the order of nature originally appointed by him; but because he governs heaven and earth by his providence, and regulates all things in such a manner that nothing happens but according to his counsel . . . whereas the faithful should . . . encourage themselves in adversity with this consolation, that they suffer no affliction, but by the ordination and command of God, because they are under his hand. (Calvin 1921, p. 185)

Whatever the weaknesses of Sterba's case against the compatibility of the world's evils and the existence of a God who is wholly good and all-powerful, he makes a powerful case for the conclusion that the world is not as we would expect if such a God occupied the role of sovereign authority. I would go so far as to argue that Sterba has shown, using the model of the ideally just political state, that if God did occupy such a role, God would not qualify as morally good in anything like the sense of "morally good" we would apply to such a state. Furthermore, given the obvious good consequences of a perfect God operating as supreme governing authority, teleological considerations would speak in favor of God taking up that role. Thus, it would only be by virtue of some powerful deontological constraint against doing so that God would refrain.

Given these points, I would argue that in light of Sterba's arguments, theists should hope that some deontological features of divine goodness (if not the ones sketched out here, then other ones) preclude God from intervening in the affairs of the world in the manner of a sovereign authority. By implication, they should hope that the doctrine of divine sovereignty is false. Because the alternative may be to deny either that God is all-powerful, or that God is good in anything resembling what we ordinarily mean by that term.

**Funding:** This research received no funding.

**Data Availability Statement:** Not applicable.

**Conflicts of Interest:** The author declares no conflict of interest.

## Notes

1    This is essentially what Swinburne (1998, p. 11) means by his term "efficacious freedom".

2    My formulation of this distinction is my own. It attempts to capture a crucial distinction between different ways of envisioning the relationship between moral obligations and the promotion of the good. I trust that it is mostly consonant with such articulations of the distinction between teleology and deontology as those found in Rawls (1999, pp. 21–22), Williams (1985, pp. 16) and Scheffler (1992, pp. 42–43), but the distinction as I articulate it here is specifically intended to capture the distinctive way of thinking about the Free Will Defense that Sterba exemplifies—and the alternative that is thus excluded—rather than to comprehensively capture how ethicists have understood this distinction.

3    Reitan (2000) formulates a point similar to the one made in this section, but in terms of the distinction between consequentialist and deontological approaches. I choose the current language to avoid the confusion that the "consequentialist" label is in danger of evoking.

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
