# Peer review of "Divine Omnipotence, Divine Sovereignty and Moral Constraints on the Prevention of Evil: A Reply to Sterba"

_religions, doi:10.3390/rel13090813_

Round 1
Reviewer 1 Report
The paper is well organized and well written. It raises an important challenge to Sterba's argument: that God's unlimited redemptive powers, instead of serving as potential justification for His choice not to prevent evil-causing free acts (an omission that allegedly calls for justification), serve instead to preempt any potential justification for his choice to prevent evil-causing free acts (a commission that calls for justification). My sense is that the author has adequately presented the challenge and made a persuasive initial case for it.
Author Response
I appreciate the positive assessment. Some changes have been made in light of another reviewer's remarks, in the form of adding one additional section to the paper that maps the objections to Sterba more explicitly onto an overview of his argument formulated in terms of his Moral Evil Prevention Requirements.
Reviewer 2 Report
The submitted paper suggests an interesting reply to James Sterba's monograph "Is a good God logically possible?". The core idea of the proposed objection is that there is an alternative account of Pauline Principle underlying Sterba's treatment of God's goodness. The author following Sterba places the discussion in the context of Free Will Defense (FWD) introducing teleological and deontological formulation of the FWD and corresponding formulations of Pauline Principle. This is undoubtedly the strongest part of the paper as it makes to my mind the sagnificant contribution to the current discussion.However there are several drawbacks, thinking of which might improve some logical moves in the paper. First, I suppose it would be reasonable to explicate Sterba's logical argument from evil in a more detail. It's worth doing since a reader might not understand what place occupies the point under discussion in the whole Sterba's argumentation. Moreover, Sterba's version of logical argument from evil shares with other versions the same presupposions (for instance, that the good must always eliminate the evil). And a reader might wonder wether proposed argument places difficulties only for specific Sterba's version or for all versions of logical argument from evil (so that it is a generic counterargument). Second, the paper leaves open the question of what conception of moral justification takes the author. He/she distiguishes between teleological and deontological fromulation of FWD and interpretation of Pauline Principle. Does the last fact mean that the author also sees two ways of justification of moral acts—teleological and deontological? It is not clear from the paper, while this question is substantive, for the problem for Sterba's account raised in the paper lyes according to the author directly in the interpretation of Pauline Principle. If the abouve reasoning is correct, then can we conclude that the whole difference between author's and Sterba's positions is that they take different accounts of moral justification? And than the problem for Sterba is in justifying his account of moral justification.
I do not think that these two points are crucial for the paper, but it is worth mentioning of them in the comments, perhaps.
Author Response
I have added a new section to the paper--what is now section 7--to address the concerns. In this section I offer a formulation of Sterba's proposed logical argument in terms of the Moral Evil Prevention Requirements he develops, and I then show how the arguments of this paper show that, if the proposed deontological moral perspective is adopted, MEPRs II-III are not violated, and MEPR I ought to be either be revised or interpreted in such a way that it is not violated. It should also be clear in this section, if it was not sufficiently clear in earlier sections, that the perspective used to challenge Sterba adopts a deontological account of justification.
Reviewer 3 Report
This is an excellent paper, clearly written and convincingly argued. The only claims that are not well-supported are the (in my opinion overly generous) claims at the last two paragraphs of the paper asserting, in effect, that Sterba has successfully shown that horrific evil is incompatible with the existence of an omnipotent and wholly good God who occupies the role of absolute sovereign. You say that you would go so far as to argue for this, but I see no good reason to go that far. There are other plausible objections to Sterba's argument besides the ones beautifully developed in this paper.
The paper needs to be proof read. There are numerous typos, e.g., on lines 34, 353, 472, 583, and 696.
Author Response
I appreciate the positive assessment. Some changes have been made in light of another reviewer's remarks, in the form of adding one additional section to the paper that maps the objections to Sterba more explicitly onto an overview of his argument formulated in terms of his Moral Evil Prevention Requirements. I have not changed the remarks at the end that strike you as overly generous. I am of course open to being shown that this is the case (perhaps in terms of other criticisms of Sterba), but insofar as this goes beyond the primary aims of the paper and is an expression of how I happen to see things at this stage, I thought I would keep it as is. Perhaps it will spark scholarly discussion of whether the evils of the world, aside from speaking to the existence of God, speak to divine sovereignty.